# Upgraded User-Friendly Image-Activated Microfluidic Cell Sorter Using an Optimized and Fast Deep Learning Algorithm

**DOI:** 10.3390/mi13122105

**Published:** 2022-11-29

**Authors:** Keondo Lee, Seong-Eun Kim, Seokho Nam, Junsang Doh, Wan Kyun Chung

**Affiliations:** 1Department of Mechanical Engineering, Pohang University of Science and Technology, Pohang 37673, Republic of Korea; 2Department of Materials Science and Engineering, Seoul National University, Seoul 08826, Republic of Korea

**Keywords:** Microfluidic Flow Cytometry, image-based cell sorting, deep learning

## Abstract

Image-based cell sorting is essential in biological and biomedical research. The sorted cells can be used for downstream analysis to expand our knowledge of cell-to-cell differences. We previously demonstrated a user-friendly image-activated microfluidic cell sorting technique using an optimized and fast deep learning algorithm. Real-time isolation of cells was carried out using this technique with an inverted microscope. In this study, we devised a recently upgraded sorting system. The cell sorting techniques shown on the microscope were implemented as a real system. Several new features were added to make it easier for the users to conduct the real-time sorting of cells or particles. The newly added features are as follows: (1) a high-resolution linear piezo-stage is used to obtain in-focus images of the fast-flowing cells; (2) an LED strobe light was incorporated to minimize the motion blur of fast-flowing cells; and (3) a vertical syringe pump setup was used to prevent the cell sedimentation. The sorting performance of the upgraded system was demonstrated through the real-time sorting of fluorescent polystyrene beads. The sorter achieved a 99.4% sorting purity for 15 μm and 10 μm beads with an average throughput of 22.1 events per second (eps).

## 1. Introduction

The isolation of interested cells from the heterogeneous populations is essential in biological and biomedical research [1,2,3,4,5,6,7,8,9,10]. The sorted cells can subsequently be used for performing downstream analysis, such as single-cell analysis. It is significant in broadening our knowledge of the cell-to-cell differences [1,4,5].

Conventional cell isolation techniques, for example, fluorescence-activated cell sorting (FACS), rely on the light scattering intensity and fluorescent bio-markers. These techniques offer several limitations [1,4,11]. Scattered light signals from the forward- and side-detectors of the FACS lack spatial information. It is difficult to obtain detailed information such as the location of the bio-marker and morphological features using the conventional techniques [1,11]. In addition, the fluorescent bio-marker can lead to random changes in the property of the isolated cells, thus affecting the downstream analysis [4].

The advent of image-based cell isolation techniques addresses the limitations of the conventional cell isolation techniques by providing information-rich cell images [1,2,3,4,5,6,7,12]. Moreover, integrating the deep learning networks with high throughput single-cell images obtained from the image-based cell sorting techniques enables the intelligent selection of the target cells from the large heterogeneous mixtures [2,5,6,13].

However, the development of a deep learning-enabled image-based FACS-like sorting system is challenging because of the short span (∼200 μs) for which the cell is available in the sorting system [12]. The flowing cell’s transit time between the interrogation and sort region can be increased from hundreds of microseconds to tens of milliseconds by introducing a microfluidic platform into the FACS-like sorting system [12]. The microfluidic cell sorting system is beneficial as it provides enough image reconstruction and processing time, to come up with a sort decision before the cell arrives at the sort region [5,12]. Increasing the flow speed of the cell aids in improving the sorting purity or throughput of the cell sorting system. A flow speed greater than 1 m/s can lead to low imaging sensitivity or motion blur issues [14]. Higher flow speeds fail to provide sufficient time for data processing, such as the image classification of deep learning algorithm [11]. Therefore, several advanced imaging methods such as frequency-division multiplexed microscopy [5] and virtual-freezing fluorescence imaging [6,14] have been employed to resolve the motion blur issue of the fast-flowing cell. Novel deep learning pipelines have also been implemented to reduce the computation time by avoiding the computationally expensive signal processing and feature extraction steps [11]. Additionally, implementing a deep learning network under the TensorRT framework [15] facilitates fast computations.

We previously demonstrated a user-friendly image-activated microfluidic cell sorting technique using an optimized and fast deep learning algorithm [15]. Three-dimensional (3D) hydrodynamically focused cells flowing through the microfluidic channel are detected, tracked, and classified in real-time using a conventional image processing pipeline and a deep learning network accelerated under the TensorRT framework. This entire process is executed within 3 ms. In addition, a user-friendly custom-written GUI program and a syringe-driven cell sorting device help users easily perform the sorting experiments [15].

In this paper, we report on the upgraded user-friendly image-activated microfluidic cell sorter using an optimized and fast deep learning algorithm. The three newly added features are as listed hereafter. (1) A high-resolution linear piezo-stage to obtain focused cell images, (2) an LED strobe light to acquire the blur-free image of fast-flowing cells, and (3) a vertical syringe pump setup to overcome the cell sedimentation. The obtained results can afford solutions to the practical problems related to cell sorting experiments.

## 2. Materials and Methods

### 2.1. Microfluidic Device

Polydimethylsiloxane (PDMS) (Slygard184, Dow Corning, USA) sorting chip was fabricated using soft lithography. The 5:1 PDMS mold was cured at 80 ${}^{\circ }$C for 4 h before plasma bonding to increase the stiffness and hardness of the sorting chip [16]. Subsequently, O${}_{2}$ plasma treatment of PDMS was performed to produce a permanent and irreversible bond between the PDMS and glass (S9213, Matsunami, Japan). The fabricated sorting chip consists of three regions: flow-focusing, detection, and sorting. In the flow-focusing region, there are five inlets (width, 100 μm) for a sample flow and four sheath flows. Thus, the sample flow is vertically and horizontally focused by four sheath flows in this region. Moreover, filters are added to all inlets to prevent the clogging of microchannels by the sample. The detection region is located in the middle of the focusing and sorting regions and is 970 μm away from the actuation channel. The imaging source camera observes cells flowing through this detection region in the main channel (width, 100 μm). In the sorting region, there are an actuation channel (width, 100 μm), a waste channel (width, 80 μm), and a collection channel (width, 40 μm). Due to the asymmetric structure of the collection and waste channels, cells flow out to the waste channel. Conversely, when a sorting signal is generated, the pushing force is transmitted from the actuation channel, and the cell is forced to flow out to the collection channel. The Photron camera observes this sorting region to monitor the sorting process. The channel height is 32 μm, which is enough for it to avoid interference with the sample flow (Figure 1).

### 2.2. Bead Sample Preparation and Loading

A mixture of polystyrene beads was prepared for evaluating the real-time sorting performance of the upgraded cell sorter. 15 μm and 10 μm fluorescent polystyrene bead solutions (F8837 10 mL and F8836 10 mL, Thermo Fisher Scientific, USA) were mixed at a 1:2 volume ratio, resulting in a pre-sorting ratio of 14.9% and a concentration of 2.73 × 10${}^{6}$ particles/mL. The bead mixture and deionized water were loaded into 1 mL and 10 mL plastic syringes for the sample and sheath fluids, respectively (BD Luer-Lok^TM^ 1 mL Syringe and 10 mL Syringe, BD Plastics, UK). The sample and sheath fluids are injected into the microfluidic sorting device using three syringe pumps at constant flow rates (sample: sheath1: sheath2 = 2:6:22 μL/min).

### 2.3. Image Processing Pipeline for the Real-Time Sorting

The image processing pipeline consists of three threads: the camera thread, the image processing thread, and the classification thread. The camera thread continuously receives 10×-magnification unsigned 8-bit 720 × 112 × 3-pixel images at 2000 fps from the Imaging Source camera (DFK 37BUX287, Imaging Source, Germany) and sends them to the image processing thread with a capture time. Then, in the image processing thread, a background image is subtracted from the acquired microscopic image, such that only cells remain in the image. A region of interest (ROI) is established for a microchannel to reduce the processing time. Upon the completion of this step, a Gaussian blur is applied to remove the noise in the given image, and then thresholding generates a binary image of the cells. The application of the morphological opening to the binary image removes noise. Contours are located to identify each cell in the image, and then the cell images are cropped from the original color image. These cropped images are used to determine the cell types through a convolutional neural network (CNN) with NVIDIA TensorRT implementation in a classification thread, and the coordinates of the centroids of the cells are used for multiple cell tracking with a hard-coded tracking algorithm. A bounding box and trajectory for each cell are drawn on an original color image.

### 2.4. Data Preparation for Training a CNN

To prepare the training dataset for the convolutional neural network (CNN), unsigned 8-bit 50 × 50 × 3-pixel images of fluorescent polystyrene beads of 15 μm and 10 μm were acquired using the cell sorting program under various lighting and focus conditions. This helped in building a robust CNN. Seventy thousand images were prepared for each class. Further, they were randomly mixed, and later divided into training (80%), test (10%), and validation (10%) sets. ResNet18 [17] coded with Keras API of TensorFlow 2.6.0 on Python 3.7.10, CUDA Toolkit 11.1, and cuDNN 8.1.0 was used as a CNN for image classification. It was trained on the prepared dataset for 50 epochs with an Adam optimizer (learning rate = e${}^{-5}$). The trained CNN was reimplemented under the TensorRT framework 7.2.1.6 to accelerate the image classification by several times.

### 2.5. Upgrade of Cell Sorting System

The real-time cell sorting mechanism used in the upgraded system (Figure 1) is identical to a previously reported image-based cell sorting technique [15]. Instead, the technique shown on the microscope was implemented as a single system on an aluminum breadboard (Figure 2). The following features have been changed or newly added to develop a more user-friendly sorting system that solves sedimentation problems and many such issues. (1) A high-resolution linear piezo-stage (SLC-2490-D-S, SmarAct, Germany) was installed to obtain focused cell images while precisely adjusting the position of the 10× objective lens (UPlanFLN, NA = 0.30). (2) LED strobe light (Constellation 120E, Veritas, USA) was used to acquire the blur-free image of the fast-flowing cells. (3) The syringe pump setup (Fusion 200 Touch, REVODIX, South Korea) was installed vertically to overcome the particle sedimentation. (4) Three XYZ-stages (XYZLNG60, MISUMI, Japan) were used to manually adjust the position of the Imaging Source camera, Photron camera (IDPExpress R2000, Photron, Japan), and syringe-driven sorting device. (5) In the syringe-driven sorting device, a 90 mm long Tygon tube (ND-100-80, Cole-Parmer, USA) was used instead of a long stainless needle, and the 90 μm piezoelectric actuator was replaced by a 30 μm piezoelectric actuator (P-840.20, PI, USA) with a shorter rising time of 0.383 ms at 100 V (Appendix A). An Anti-vibration pad (AVP-3, REFCO Manufacturing Ltd., Switzerland) was attached to the bottom of the XYZ-stage to prevent the vibrations from getting transmitted to the sorting chip, when the piezoelectric actuator expands or contracts. (6) A 3D-printed tube holder was made to ease the collection of the sorted and unsorted cells from a sorting chip.

## 3. Results and Discussion

### 3.1. Processing Time and Classification Accuracy of the Upgraded Cell Sorting System

The upgraded system uses a higher performance multi-core processor (i9-10900KF, Intel, USA) and graphic processing unit (GeForce RTX 3090, NVIDIA, USA) as compared to the previous system [15]. This resulted in reduced processing and inference time. 99.3% of the entire image processing process ended in less than 2 ms, while the average image processing and inference times are 0.16 ms and 0.43 ms, respectively (Figure 3). The maximum detection throughput of the upgraded system is improved to ∼2000 events per second (eps). Moreover, the classification accuracy of the upgraded system for the polystyrene bead image is 99.9% (Appendix A).

### 3.2. High-Resolution Linear Piezo-Stage and LED Strobe Light to Acquire In-Focus Blur-Free Images of the Fast-Flowing Cells

Acquiring in-focus cell images is very important to separate the target cells from two samples of similar cell sizes with a high sorting purity. For the same cell, different cell images were taken by changing the distance between the objective lens and the 3D-focused flowing cells (Appendix A). Furthermore, manually maintaining the same focus condition during the training data collection and sorting processes is extremely difficult. Further, the classification accuracy of the CNN significantly decreases if the cell images are taken under different focus conditions. In the previous version of the cell sorting system, the objective lens of an inverted microscope was manually manipulated to focus a microscope. Thus, in the case of similar-sized cells, separating target cells with a high sorting purity was challenging. To overcome this limitation and to make the system more user-friendly, a high-resolution linear piezo-stage was installed under the objective lens in the newly upgraded system (Figure 4A). The user could move the objective lens with a nanometer resolution using the custom C++ cell sorting program.

Acquiring blur-free images of the fast-flowing cells without sacrificing imaging sensitivity is also important and challenging. In the previous system, the blur-free cell images were obtained by decreasing the exposure time of the Imaging Source camera. This leads to images with low sensitivity. Conversely, in the upgraded system, microscopic images of fast-flowing cells were obtained by using strong LED strobe light with a duration of 10 μs synced with the Imaging Source camera, resulting in motion-blur-free cell images (Figure 4B). Considering the pixel size of the Imaging Source camera as 6.9 μm, a maximum allowable flow speed of 0.69 m/s is obtained by dividing the camera’s pixel size with the exposure time (10 μs). This helped obtain a blur-free microscopic image of the fast-flowing cells [14]. By further lowering the exposure time to 1 μs, which is the maximum performance of the Imaging Source camera, a motion-blur-free image of cells moving faster than 1 m/s can be obtained. The images in Figure 5 show the obtained blur-free cell images using the upgraded system for different exposure times.

### 3.3. Vertical Syringe Pump Setup to Prevent Particle Sedimentation

In the previous system, the sorting throughput rapidly decreased over time owing to particle sedimentation. The sedimentation velocity (V) is determined by Stokes’ law ($V=\frac{\left(\rho_{p}-\rho_{f}\right)gD^{2}}{18\mu }$), where $\rho_{p}$ is the density of the sphere, $\rho_{f}$ is the density of the fluid, g is the acceleration due to gravity, D is the diameter of the particle, and μ is the dynamic viscosity of the fluid [18]. Common ways to prevent sedimentation are (1) to spin a magnetic stir bar (immersed in a sample fluid) with high velocity by using a small motor [19], (2) to use a biocompatible viscous carrier fluid to decrease the sedimentation velocity [20], and (3) to rotate the syringe in order to retain the particles in a suspended state [18]. Methods 1 and 3 are futile if the tube length is long and all beads sink before they reach the inlet of the microfluidic chip. According to Stokes’ law calculation, it takes 2 min and 30 s for 10 μm polystyrene beads to sink inside a Tygon tube with an inner diameter of 0.51 mm. When the sample flow rate is 2 μL/min, the Tygon tube length from the syringe needle tip to the sample inlet should be less than 24.3 mm in order to prevent particle sedimentation. If larger beads (greater than 10 μm beads) are used, the tube length should be much shorter as they sink faster. It has been observed that shortening the tube length and attaching the syringe pump close to the sorting chip leads to problems in the sorting experiments. In contrast, the second method is comparatively better. This method prevents the sedimentation by increasing the viscosity of the carrier fluid in order to increase the sedimentation time. For example, OptiPrep^TM^, a biocompatible density gradient medium, can be used to prevent cell sedimentation [20]. However, when OptiPrep^TM^ is mixed with a sample fluid, it was observed that clear images of the hydrodynamically focused sample stream were obtained by increasing the concentration of OptiPrep^TM^ (Appendix A). This leads to a reduced classification accuracy of the deep learning network because the focused stream clearly appears in the cell image and may cover the cells.

Therefore, in the upgraded system, the syringe pump was set vertically right above the sorting chip. Thus, the Tygon tube was set vertically straight to avoid particle sedimentation without using OptiPrep^TM^. During the five consecutive bead sorting experiments, the throughput decreased slowly at about 20 eps (Appendix A). This observation is somewhat consistent with the Stokes’ law calculation that it would take 1 h 37 min for 10 μm beads to sink in a 1 mL plastic syringe containing 0.2 mL sample fluid.

### 3.4. Real-Time Sorting of Fluorescent Polystyrene Beads

Real-time sorting of fluorescent polystyrene beads was conducted to evaluate the performance of the upgraded system. The 15 μm target beads (blue) were sorted into the collection channel, and the 10 μm non-target beads (yellow-green) were sorted into the waste channel. Each sorting experiment was performed for 20 min, and a total of five consecutive experiments were performed (Appendix A). The sorting throughput was maintained between 20 and 30 eps during the experiment. The throughput can be raised further at the expense of sorting purity and yield [6].

The sorting results were analyzed by manually counting the numbers of sorted and unsorted beads using a fluorescence microscope (Appendix A). According to the formula [15], the pre-sorting ratio, post-sorting ratio (sorting purity), yield, and enrichment factor were calculated as 14.3%, 99.4%, 94.4%, and 6.67, respectively. Figure 6 show the fluorescent images of the pre-sorting, post-sorting, and waste.

## 4. Conclusions

Our previous user-friendly image-activated microfluidic cell sorting technique successfully demonstrated the real-time sorting of beads and cells with a high sorting purity. We herein present a recently upgraded user-friendly image-activated cell sorting system. A high-resolution linear piezo-stage and LED strobe light were newly added to the upgraded system in order to obtain in-focus blur-free images of the fast-flowing cells. In addition, a syringe pump for sample injection was vertically installed to prevent the particle sedimentation. Several experiments confirmed that the sorting throughput of the upgraded system slowly decreased over time. The proposed system can be easily implemented using commercially available items. However, the moving speed of the flowing cells is limited to less than half of the horizontal field of view divided by the frame interval to measure the moving speed of cells using a CMOS camera. Real-time bead sorting also demonstrated that the upgraded system achieved 99.4% sorting purity and 94.4% yield for 15 μm and 10 μm beads with an average throughput of 22.1 eps. Our findings will potentially be helpful in handling practical problems such as motion blur and particle sedimentation. In addition, the upgraded system can be utilized in various applications, including biological research and therapy.

## Figures and Tables

**Figure 1 micromachines-13-02105-f001:**
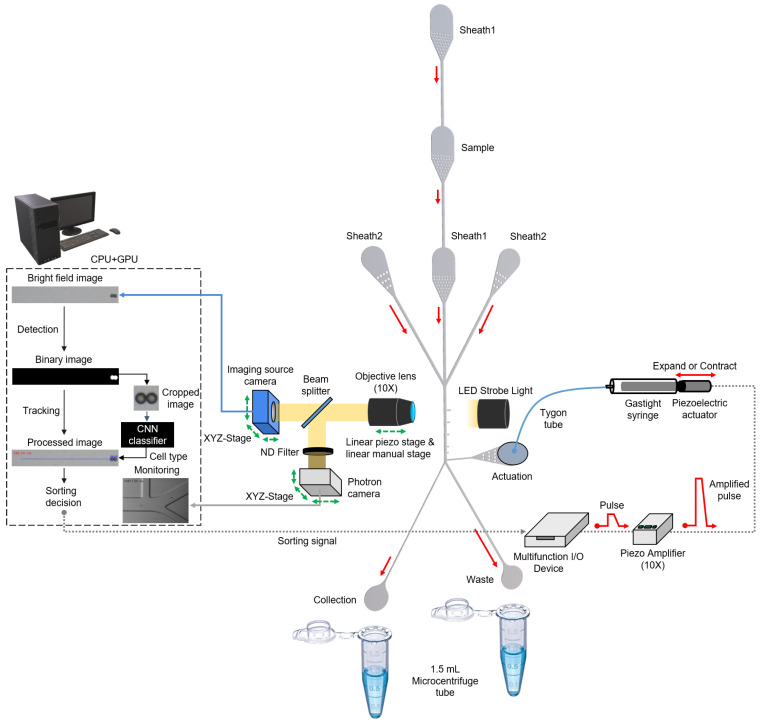
Schematic of upgraded user-friendly image-activated microfluidic cell sorter. The real-time sorting mechanism has not changed significantly compared to the previous system [15]. The vertically and horizontally focused cells are analyzed as they flow through the detection region. Target and non-target cells are sorted into the collection and waste channels based on the classified cell type.

**Figure 2 micromachines-13-02105-f002:**
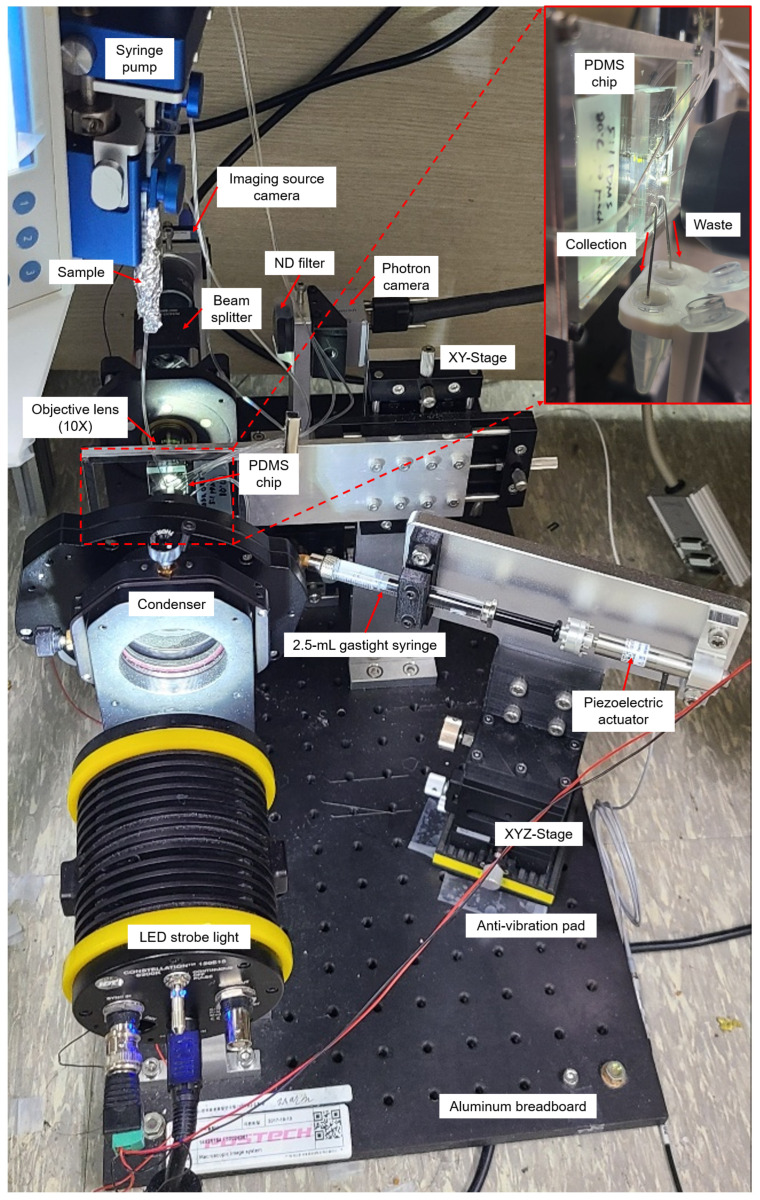
Upgraded user-friendly image-activated microfluidic cell sorter. In the upgraded system, the previously reported sorting technique [15] was implemented so that the cells could be sorted without cell sedimentation.

**Figure 3 micromachines-13-02105-f003:**
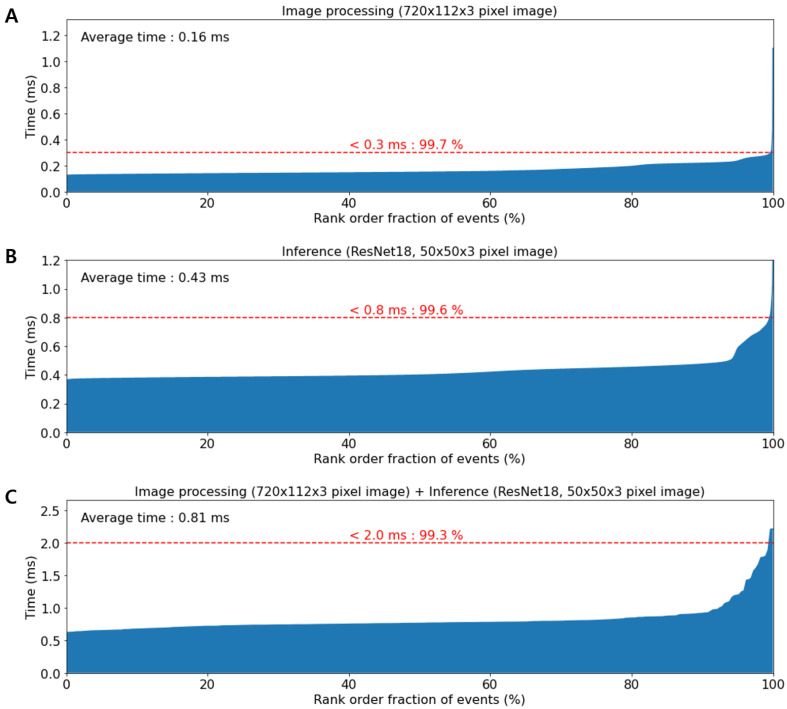
Plots of computation time. (**A**) Image processing time without image classification. The average time of image processing is 0.16 ms. (**B**) The inference time on ResNet18 with 50 × 50 × 3-pixel images. The average time of image classification is 0.43 ms. (**C**) The total processing time. The graph indicates that 99.3% of total processing was completed within 2.0 ms.

**Figure 4 micromachines-13-02105-f004:**
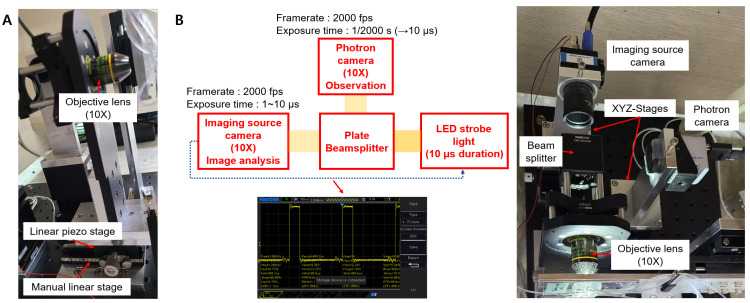
(**A**) High-resolution linear piezo-stage to obtain in-focus images of the flowing cells, (**B**) LED strobe light to minimize the motion blur issue of the fast-flowing cells.

**Figure 5 micromachines-13-02105-f005:**
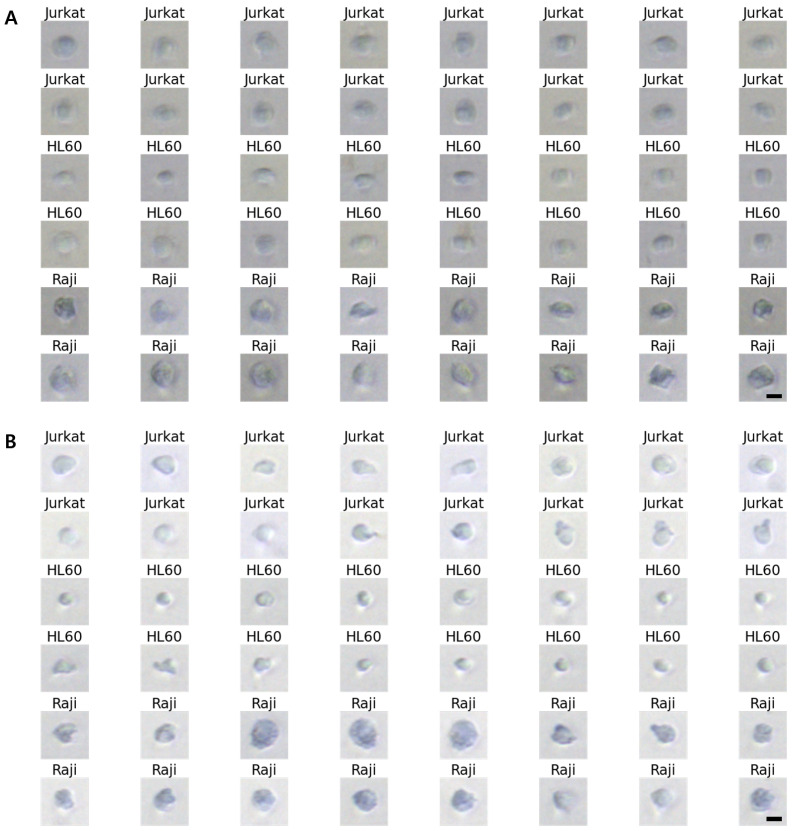
10× magnified Differential Interference Contrast (DIC) images of Jurkat, HL60, and Raji cells obtained using the upgraded system. (**A**) exposure time = 2 μs, (**B**) exposure time = 4 μs. Scale bars, 10 μm.

**Figure 6 micromachines-13-02105-f006:**
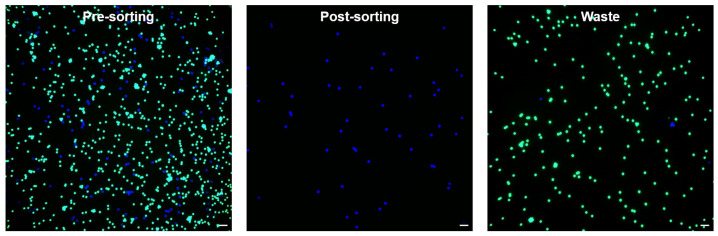
Fluorescence microscopy images of bead sorting performed on the upgraded system (scale bars 100 μm). The blue and green dots indicate the target 15 μm bead and the non-target 10 μm bead, respectively.

## Data Availability

The datasets generated during and/or analyzed during the current study are available from the corresponding author on reasonable request.

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
