# Peer review of "Upgraded User-Friendly Image-Activated Microfluidic Cell Sorter Using an Optimized and Fast Deep Learning Algorithm"

_micromachines, 2022, doi:10.3390/mi13122105_

Round 1
Reviewer 1 Report
The manuscript entitled "Upgraded user-friendly image-activated microfluidic cell sorter using an optimized and fast deep learning algorithm” shows the improved system ensuring high throughput cells sorting based on intelligent image processing. The work can be vital to the biomedical fields, since fast and reliable cells classification can be necessary for the preliminary bio-analysis in the context of cells quality, size, shape, morphology, etc. Although the scope is interesting, and the results are promising, many aspects have to be discussed before the manuscript can be published.
Major comments:
1) Even though the article describes upgraded system and refers a lot the previously published article, there should be included the scheme and real view of the microfluidic structure used
2) Why PDMS was mixed in atypical 5:1 proportion?
3) Could you explain all the steps for the image processing time which is less than 2 ms)? Only processing and inference time are mentioned (0.16 ms and 0.43 ms). The rest of the time is for?
4) All the system is rather large scale and hardly portable. Could you comment on the potential miniaturization of the components?
5) The quality of the Fig. 5 and cells images are not appropriate. Did you verify if it is sufficient for the biomedical science?
6) Line 181-183 - Could you give more explanation on the medium concentration increase, better image clearance and reduced classification accuracy relation?
7) Why the no of sorted and unsorted beads was counting manually? Why don’t you use any software?
8) Why did you check the performance of optics for real biological object (e.g. Jurkat cells) and conduct sorting only for beads?
9) The conclusion section needs to be improved. You refer only to the previous article, while overall merit is not visible. You should clearly present you system herein, mention its advantages and disadvantages, metrological outcomes, future perspective and big picture vision.
Minor comments:
1) The template for the manuscript is for “Applied Sciences”, but you are wishing to publish with “Micromachines”.
2) Could you provide the information on camera model?
3) Please, shift Figure 6 below the paragraph it is mentioned in
4) Please, check the article in the context of typos (e.g. line 149)
Author Response
We thank the editor and reviewer for their valuable comments on the paper. In response, we have revised the manuscript and addressed all points that reviewer raised. Please see the attachment.

Reviewer 2 Report
In this work, authors upgraded previously designed image-activated microfluidic cell sorter in the aspect of piezo-stage for better obtain of in-focus images, LED strobe light to minimize cell motion blur, and vertical syringe pump setup to prevent cell sedimentation. There are a few of questions that need to be addressed before can be considered for publication.
1. Overall, the novelty of the work should be clarified in the Introduction section. There has already been some work regarding developing deep learning integrated image activated cell sorter with fast throughput and more applications:
Tang, Rui, et al. "Low-latency label-free image-activated cell sorting using fast deep learning and AI inferencing." Biosensors and Bioelectronics (2022): 114865;
Isozaki, Akihiro, et al. "Intelligent image-activated cell sorting 2.0." Lab on a Chip 20.13 (2020): 2263-2273.
what is the advantage of the work in this manuscript?
2. Authors mention a high resolution linear piezo stage is used to obtain in-focus images of flowing cells, however, the improvement in the cell separation results by using this stage should be discussed in the paper. In addition, by implementing this stage, the movement accuracy of the objective lens will be surely improved, but what is the standard in determining a good cell focus condition? By naked eye or there is a system recognising best in-focus images?
3. Can the exposure time set directly from the camera? What is the direct function of the LED strobe light? Does the LED strobe light works by supply additional light intensity to improve the image sensitivity and reduce exposure time?
4. What is the separating resolution?
5. What is the maximum particle concentration the sorter can process?
6. Authors mentioned “when OptiPrepTM is mixed with a sample fluid, it was observed that clear images of the hydrodynamically focused sample stream were obtained by increasing the concentration of OptiPrepTM”, can authors explain why?
7. In Figure 5, a scale bar is needed. In addition, authors demonstrated cell images captured with different exposure time, however, how is the exposure time affect the separation results? Since images of different cells have been captured, why not demonstrate the cell separation to different outlets?
8. There are some typos and grammar mistakes. The manuscript needs to be carefully checked. In addition, the logic and flow need to be improved between sentences.
Author Response

(The authors gave the same response as above.)

Reviewer 3 Report
Manuscript ID micromachines-1995951
Title Upgraded user-friendly image-activated microfluidic cell sorter using an optimized and fast deep learning algorithm
Authors Keondo Lee , Seong-Eun Kim , Seokho Nam , Junsang Doh , Wan Kyun Chung *
The manuscript describes the development (upgrade) of a microfluidic cell-sorter platform.
The process uses high speed image activation for sorting the cells to either harvesting or to waste.
The high speed of imaging allows for high flow rate and a high cell count (high number of events per sec). The platform produces blur free images of fast flowing cells which then provides high sorting purity and high yield to the overall system.
The paper is well written and easy to read. The figures and images are well done and illustrative the work. The manuscript is a well constructed paper.
Recommendation accept as is.
Author Response
Thank you so much for taking the time to give us your feedback. We really appreciate your comments.
Round 2
Reviewer 1 Report
The Authors have responded to all the comments. I accept the manuscript in the recent form.